# Utilizing Satellite Data to Establish Rainfall Intensity-Duration-Frequency Curves for Major Cities in Iraq

Sarah Jabbar Zeri [1], Mohammed Magdy Hamed [2] , Xiaojun Wang [3,4,*] and Shamsuddin Shahid [1,*]

1 Department of Water & Environmental Engineering, Faculty of Civil Engineering, Universiti Teknologi Malaysia, Skudai 81310, Johor, Malaysia
2 Construction and Building Engineering Department, College of Engineering and Technology, Arab Academy for Science, Technology and Maritime Transport (AASTMT), B 2401 Smart Village, Giza 12577, Egypt
3 State Key Laboratory of Hydrology–Water Resources and Hydraulic Engineering, Nanjing Hydraulic Research Institute, Nanjing 210029, China
4 Research Center for Climate Change, Ministry of Water Resources, Nanjing 210029, China
* Correspondence: xjwang@nhri.cn (X.W.); sshahid@utm.my (S.S.)

**Abstract:** This study generates intensity-duration-frequency curves for three important cities in Iraq using Global Precipitation Measurement Integrated Multi-Satellite Retrievals for Global Precipitation Measurement (IMERG), Global Satellite Mapping of Precipitation near real-time (GSMaP NRT), and gauge corrected (GSMaP GC) satellite precipitation datasets. Many probability distribution functions were used to fit the maximum yearly rainfall data. The Sherman equation was used to create intensity-duration-frequency (IDF) curves for rainfall intensities with 2-, 5-, 10-, 25-, 50-, and 100-year return periods, with the estimated coefficients of the best-fit distribution serving as the fitting parameters. The discrepancy between the IDF curves produced from the satellites and the observed data was used to bias correct the satellite IDF curves. The Generalized Extreme Value Distribution model best describes the hourly rainfall distribution of satellite data. GSMaP GC was the best option for creating IDF curves with higher correlations with observed data at Baghdad, Basra, and Mosul. The study indicates the necessity of gauge correction of satellite rainfall data to reduce under- and over-estimating observed rainfall. GSMaP GC can reasonably estimate rainfall in a predominantly arid climate region like Iraq. The generated IDF curves may be an important step toward achieving sustainable urban stormwater management in the country.

**Keywords:** urban flooding; information shortage; IDF curve; stormwater management

## 1. Introduction

Flash floods are a major climatic risk in many countries of the world [1]. This is generally attributed to inadequate urban stormwater management systems (SWMS), particularly for cities in developing countries. Urbanization causes a reduction of permeable surfaces and increased runoff, which eventually alters the urban hydraulic dynamics and the dimension of rivers and streams [2]. This contributes to the increased frequency and severity of urban floods. While the number of deaths in urban floods is usually low, financial losses due to the destruction of buildings and other structures and the disruption of commerce are substantial [3,4]. Both the population and the value of assets in disaster-prone areas are growing [5]. It also severely disrupts the people's well-being and urban livelihoods. The global urban area has increased from 0.6 to 0.9 million $km^2$ from 2000 to 2010 [6]. However, the urban infrastructures are not improved proportionately with its expansion in many regions [7–9]. In some cases, it is often updated but not in line with altering rainfall patterns. Consequently, urban floods caused a dramatic increase in economic loss globally [2]. According to the Intergovernmental Panel on Climate Change (IPCC) [10], there will be a rise in the frequency of extreme rainfall events and a decrease in the frequency of low rainfall events.

The Middle East and North Africa (MENA) is one of the most vulnerable places on Earth to climatic hazards [11]. The effects of such hazards are projected to be highest in Iraq among the MENA countries [12]. As a result of decades of conflict and poor administration, Iraq is now vulnerable to any natural calamity. There has been a rise in the occurrence and severity of weather-related disasters in recent years, and their effects are becoming increasingly obvious throughout Iraq [13]. Though drought, aridity, and temperature extremes are considered the most destructive disasters in Iraq, a substantial rise in rainfall extremes and flash floods has been noticed in recent years.

Salman et al. [14] indicated changes in daily rainfall mean and variability in Iraq during 1965–2015, and a consequent decrease in heavy rainfall events in 53% of stations over Iraq. Therefore, the increase in flash floods may be linked to inadequate drainage systems with urban development [7]. Iraq has experienced rapid urbanization in the last few decades. The urban population in the country has increased from 10.53 million in 1985 to 31.48 million in 2020. The urban SWMS has also developed proportionately. However, those structures are designed based on available data for the historic period. Daily data were often employed to assume the recurrence of hourly or sub-hourly rainfall extremes. This caused the failure of SWMS and urban floods in Iraq. This is also clearly comprehensible from the accelerated impacts in urban areas. However, one of the major principles of the country's national development plan 2010–2014 is sound environmental management for sustainable urban development [15]. This underlines the need to design urban hydraulic structures based on the local rainfall extremes. Knowing the features of intense rain events, or the intensity-duration-frequency (IDF) relationship, is crucial in this regard. Sustainable urban development can be promoted by adopting guidelines for urban drainage systems based on IDF. Such curves are developed from the time series of annual peak rainfall events for different durations. It allows for estimating the recurrence of different intensities of rainfall, which is required for estimating peak runoff for designing hydraulic structures [16,17].

Despite its critical importance, very little research has been conducted on the required rainfall properties for urban hydraulic infrastructure. This is partly because of the difficulty of undertaking such research without access to high-resolution hourly or sub-hourly rainfall data. A long-term hourly precipitation record is required for IDF curve generation. There is a severe lack of such data in Iraq as the country's meteorological stations have been out of service for a long time. In addition, several stations had been damaged due to the military actions in the recent conflicts. Without a dense rainfall monitoring network or better temporal resolution (hourly or sub-hourly data), hydrological researchers have increasingly turned to satellite precipitation data in recent years. Satellite rainfall data has been utilized in recent years to estimate drought in Iraq [18–21]. Some studies also showed the reasonable functioning of satellite rainfall products in estimating rainfall extremes in Iraq [22,23]. Unfortunately, such data has not been used to characterize rainfall extremes for deriving urban drainage design parameters in Iraq.

Several studies have used satellite rainfall data to build IDF relationships in an ungauged area [3,24–27]. The biggest issue with using satellite data to create IDF curves is that they tend to underestimate extreme rainfall occurrences [28–30]. Noor et al. [28] indicated that all four remote-sensing rainfall methods significantly underestimate the rainfall intensities in Peninsular Malaysia throughout a range of durations and return periods. Nashwan et al. [31] compared the accuracy of five satellite precipitation products over Egypt and showed poor performance of all products in estimating rainfall over a predominantly arid climate region like Egypt. Ziarh et al. [32] showed bias in satellite precipitation varies with topography, cloud type, and rainfall intensity. Therefore, it is recommended that in situ data be used to adjust for bias in satellite rainfall data before their use in IDF curve generation. For instance, Kyaw et al. [33] bias-corrected remote sensing data using short-period daily rainfall data available at a single location in Yangon and then employed the corrected data for generating IDF curves in nearby regions.

In this study, we employed remotely detected precipitation data for estimating IDF curves for three major cities of Iraq. Performance of three remote sensing rainfall products, Integrated Multi-Satellite Retrievals for Global Precipitation Measurement (IMERG), Global Satellite Mapping of Precipitation (GSMaP) Near Real-Time (GSMaP NRT), and GSMaP gauge corrected (GSMaP GC) were evaluated to find the best for generating IDF curves for Baghdad, Mosul, and Basrah. Updating SWMS and reducing rising urban floods in Iraq is challenging due to the lack of updated IDF curves based on the current rainfall pattern. The generated IDF curves may be an important step toward achieving sustainable urban SWM in the country.

## 2. Study Area and Data

### 2.1. Physical and Climatological Properties of Iraq

Iraq is located in southwest Asia between the coordinates 28 to 38° N and 38 to 48.5° E and has a total land area of 438,320 km$^2$ [34]. Baghdad, Mosul, and Basrah are the major Iraqi cities considered for IDF curve development in this study. The site of the cities on the map of Iraq is displayed in Figure 1, which was created using QGIS open source software. Iraq's boundary and the digital elevation model (DEM) data were downloaded from the publicly accessible DIVA_GIS website (https://www.diva-gis.org (accessed on 7 November 2022)). The most rainfall in Iraq occurs in January, while July is the highest rainfall month in other areas. The average precipitation gradually rises from the south to the north (Figure 2). However, western Iraq receives only sporadic rainfall each year [35]. Baghdad, the capital of Iraq, has an average annual precipitation of around 151.8 mm. Located in flat plains beside the Tigris River, it is home to nearly 6 million people. Basrah is Iraq's southernmost governorate, bordering Iran, Kuwait, and Saudi Arabia, with an area of 19,070 km$^2$. The city has a hot and dry climate, where summer temperatures are among the hottest in the world. Humidity and rainfall are relatively high due to the proximity to the Persian Gulf. The city gets an average of 152 mm of rain between October and May [36]. Mosul is located 362 km northwest of Baghdad. It is located near the northern highlands and receives a higher rainfall (~363.6 mm) than Baghdad and Basrah [37]. A list of extreme rainfall-driven flash floods in Iraq in the last decade is provided in Table 1. It shows an apparent increase in extreme rainfall and flooding in Iraq recently.

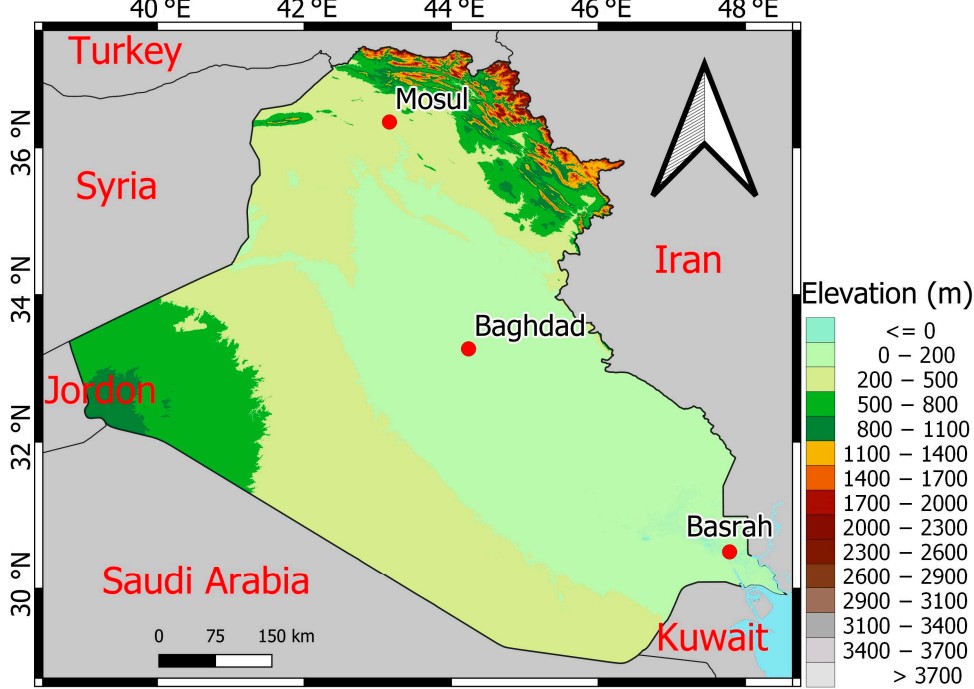

**Figure 1.** Topography of the study region.

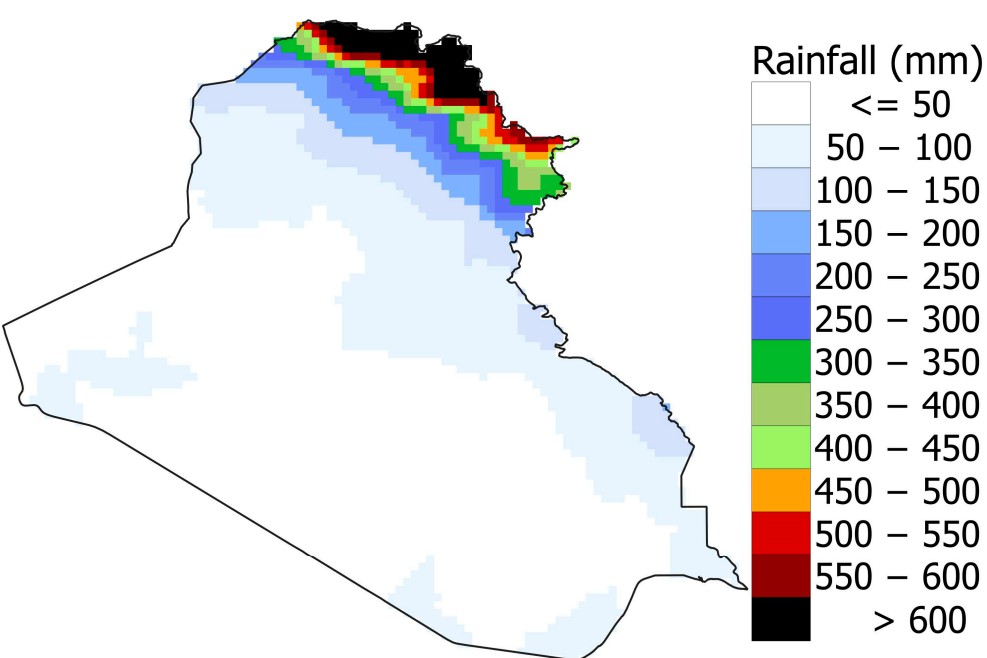

**Figure 2.** Annual mean rainfall in mm.

**Table 1.** A list of recent extreme rainfall-driven flash floods in Iraq.

| Extreme Rainfall and Floods | Date | Impact |
|---|---|---|
| Heavy rainfall and flash flood in central Iraq | 11 November 2013 | 11 death and heavy damages to building and civil structures |
| Flash flood in Baghdad | 28 October 2015 | 58 people died, and 84,000 were evacuated. Emergency was declared |
| Heavy rainfall driven flood in south Iraq | 5 May 2019 | 20,00 people were evacuated, and a hundred thousand were out of water supply |
| Heavy rainfall driven flood in Erbil | 17 December 2021 | 14 people died, and 7000 evacuated. Damage to residential buildings, infrastructure, and vehicles |
| Extreme rainfall in northern Iraq | 30 October 2021 | 3180 people were evacuated. Damages to roads |
| Heavy storms throughout the country | 24 March 2019 | 1173 families have been displaced |
| Heavy rainfall and flash floods in southern governorates | 22 November 2018 | 21 people have died, and 180 were injured as a result of the flooding |
| Recorded 67 mm of rainfall in a day in the bordering region of Iraq | 07 November 2018 | Heavy damage to infrastructure |

*2.2. Dataset*

This study also employed three types of remotely sensed precipitation data. Table 2 details the remote sensing precipitation datasets. IMERG integrates precipitation data using the Global Precipitation Measurement (GPM) satellites. The precipitation data in all the satellite products used in this study was retrieved using a combination of multiple passive microwave and infra-red sensors [31]. IMERG has three rainfall products, early, late, and final run. The IMERG final run (FR) is the most accurate of the three precipitation modes [38]. GSMaP NRT and GSMaP GC rainfall data are collected and compiled by the Core Research for Evolutional Science and Technology (CREST) of the Japan Science and Technology Agency (JSTA) in collaboration with the Japan Aerospace Exploration Agency (JAXA) Precipitation Measuring Mission (PMM) Science Team [39–41]. The former was developed by fusing cloud movement vectors derived from infrared photos with global precipitation rates derived from passive microwave radiometers [42,43]. The latter is a by-product of GSMaP NRT, developed by correcting it with precipitation of Climate Prediction Center [44]. This study used Google Earth Engine (GEE), a cloud-based system for universe geospatial research, to retrieve satellite precipitation data [45]. This platform

provides satellite precipitation products and the required tools for downloading them for a specific area or point. This study downloaded the satellite rainfall datasets of the grid locations representing the cities of Iraq using GEE.

**Table 2.** Dataset list used in this research.

| Data | Time | Spatial Resolution | Temporal Resolution | Reference |
|------|------|--------------------|---------------------|-----------|
| GSMaP NRT | 2000 to present | $0.1° \times 0.1°$ | 1-h | [41] |
| GSMaP GC | 2000 to present | $0.1° \times 0.1°$ | 1-h | [44] |
| IMERG | 2000 to present | $0.1° \times 0.1°$ | 30 min | [46] |

## 3. Methodology

### 3.1. Research Steps

The primary goal of this research was to create IDF curves for three major cities in Iraq and extend the method for developing IDF curves at ungauged areas using satellite rainfall. Figure 3 shows the flowchart elaborating the specific technical steps followed to fulfil the objectives. The goal was accomplished by performing the procedures listed below.

1. Estimate the maximum annual rainfall intensity (ARI) for the study period (2000–2021);
2. Determine probability distribution functions (PDFs) best fit the ARI time series;
3. Use the best-fit PDF for estimating rainfall intensity for each duration and return period;
4. Apply regression techniques to generate the IDF curves using the Sherman equation;
5. Repeat steps 1–4 to generate IDF curves for all locations using all three satellite-based precipitation datasets, repeat;
6. Quantify the discrepancy between each city's satellite and observed dataset IDF curves;
7. Select the satellite rainfall with the lowest IDF bias and correct it based on the observed dataset IDF.

### 3.2. Distribution Functions

A PDF chosen to fit the distribution of a particular satellite precipitation may not perform well with another dataset. As a result, comparing different PDFs to choose the best one is the best practice [28]. This study compared the performance of three generally employed PDFs, Generalized Extreme Value (GEV), Gumbel, and General Pareto (GP), to find the appropriate one that best fits the satellite ARI time series. Koutsoyiannis and Baloutsos [47] reported that Extreme Value (EV1) and Gumbel seem more appropriate for a short period of data, while GEV may be better for estimating a larger return period. Kastridis and Stathis [48] also showed that the length of the rainfall time series significantly influences the selection of the appropriate distribution. Considering the data period of 20 years used in this study, EV1 and Gumbel may be the more suitable. However, the present study relied on the findings of the previous studies in Iraq to select the distributions for their relative comparisons and identify the best distribution. According to earlier studies in Iraq, one of these three PDFs typically offered the best fit for the ARI time series. For instance, Majeed et al. [22] demonstrated that the Gumbel provided the best rainfall intensities in Najaf city, Iraq, for various return periods and durations. AL-Dulaimi et al. [49] claimed that Gumbel distribution was the best frequency analysis technique in Babylon City and Alluvial Fertile Zone, Iraq. Different stations in northern Iraq showed different best PDFs, including GEV, Gumble, and GP [50]. Thus, only these three PDFs were considered in the current investigation. The performance was assessed in estimating PDF parameters using the Maximum Likelihood (MLE) method based on the negative log-likelihood goodness of fit test. The negative likelihood ratio is frequently used to evaluate the effectiveness of a diagnostic test since it offers built-in strength of rule in or out probability [51]. Table 3 contains the equations for the PDFs.

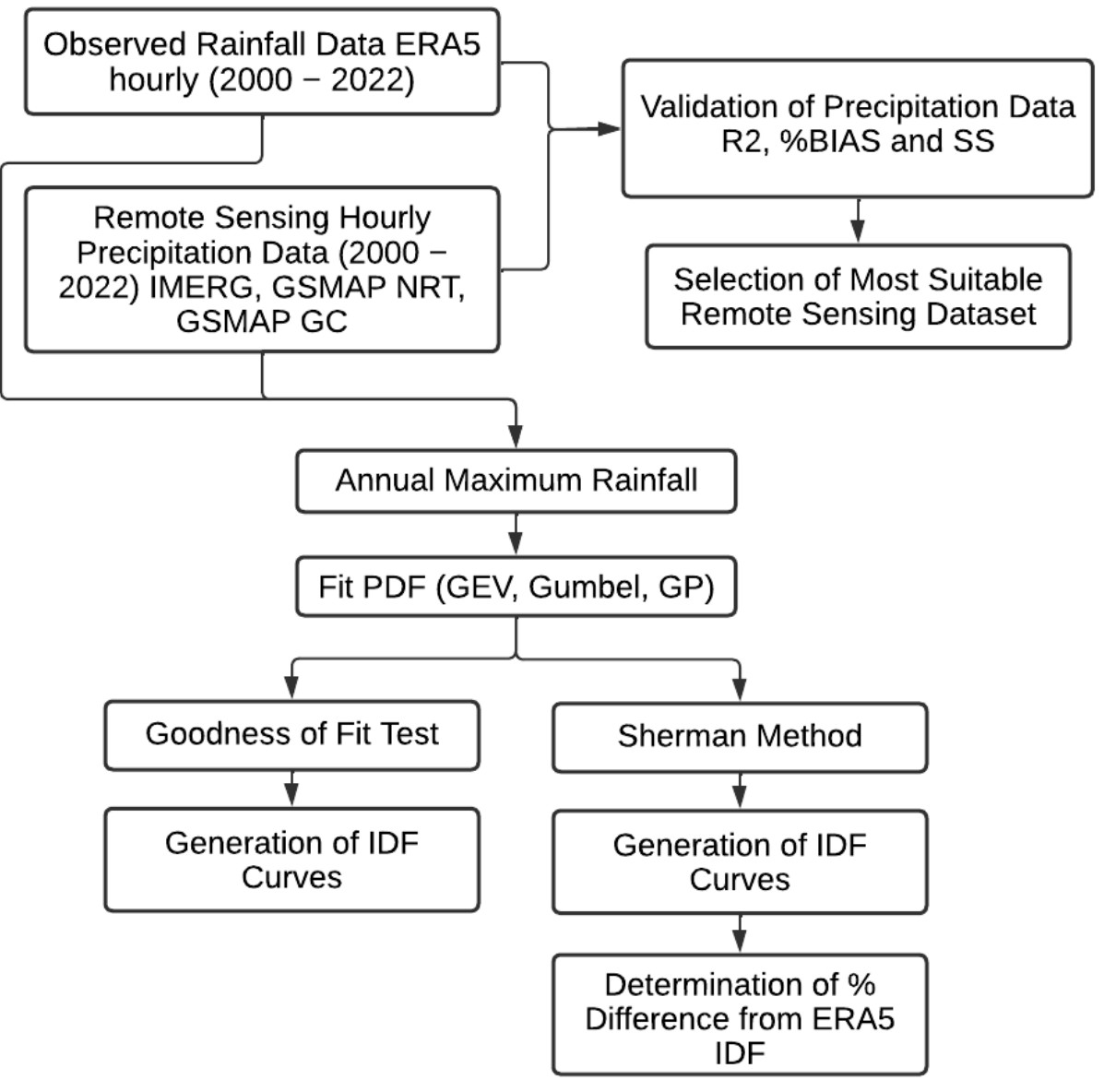

**Figure 3.** Flowchart showing the development of IDF curves.

**Table 3.** Probability distribution functions used to fit observed dataset and satellite rainfall data at different locations.

| Function | Equation | Parameter |
|---|---|---|
| GEV | $f(x) = \begin{cases} \frac{1}{\sigma} \exp\left(-(1+kz)^{-1/k}\right)(1+kz)^{-1-1/k} & k \neq 0 \\ \frac{1}{\sigma} \exp(-z - \exp(-z)) & k = 0 \end{cases}$ | $z = \frac{x-\mu}{\sigma}$ <br> $k$: shape <br> $\mu$: location <br> $\sigma$: scale |
| Gumbel | $f(x) = \frac{1}{\sigma} \exp(-z - \exp(-z))$ | |
| GP | $f(x) = \begin{cases} \frac{1}{\sigma}(1+kz)^{-1-1/k} & k \neq 0 \\ \frac{1}{\sigma} \exp(-z) & k = 0 \end{cases}$ | |

The effectiveness of the diagnostic test was evaluated using the negative log-likelihood test. [52].

$$L(\theta) = \sum_{i=1}^{n} \ln(f_i(y_i \mid \theta)) \tag{1}$$

where $y$ = likelihood function, $L(\theta)$ = log-likelihood function, and $n$ = number of observations.

### 3.3. Sherman Equation

The Sherman equation [53] was used to regress the IDF curves,

$$i = \frac{kT^x}{(t+b)^m} \tag{2}$$

where $i$ is the amount of precipitation; t is the length of the storm; $kT$ is the frequency factor; and $x$, $b$, and $m$ are the least squares-derived regression parameters. Regression was performed using various $x$, $b$, and $m$ values for different return durations.

### 3.4. Evaluation of Satellite Precipitation Data

To be employed for IDF curve estimation, satellite rainfall is expected to accurately replicate the mean, temporal pattern, and distribution of observed rainfall. The study used three robust statistical indices to measure those capabilities of satellite rainfall, Spearman coefficient of determination ($R^2$), percentage of bias (%BIAS), and Perkin's skill score (SS) [54,55]. Non-parametric Spearman $R^2$ and Perkin's skill score were used considering the high skewness of daily data. Table 4 lists the formulas, ranges, and ideal values of the indices.

**Table 4.** Statistical metrics for assessing the performance of satellite-based precipitation data.

| Index | Optimum Value |
|---|---|
| $R^2 = \frac{\sum_1^n (x_{obs,i} - \overline{x}_{obs})(x_{sim,i} - \overline{x}_{sim})}{\sqrt{\sum_{i=1}^n (x_{sim,i} - \overline{x}_{sim})^2 \sum_{i=1}^n (x_{obs,i} - \overline{x}_{obs})^2}}$ | 1 |
| $\%\text{BIAS} = 100 * \frac{\sum_{i=1}^N (x_{sim,i} - x_{obs,i})^2}{(x_{obs,i})}$ | 0 |
| $SS = \int_{-\infty}^{\infty} \min\left[pdf(x_{sim,i}), pdf(x_{obs,i})\right]$ | 1 |

## 4. Results

### 4.1. Performance of Satellite Precipitation Data

The performance of satellite precipitation products replicating observed precipitation was evaluated based on statistical and graphical metrics. Three cities in Iraq's (Table 5) indicate how well various satellite rainfall data replicated observed rainfall. The results revealed the better performance of GSMaP GC at three locations in most metrics. It was best to replicate the temporal pattern of in situ rainfall at Baghdad and Mosul, with the least bias at Baghdad and similar probability distribution at Baghdad and Mosul. GSMaP NRT showed better performance in capturing mean bias and probability distribution at Basrah, while IMERG showed the best performance in terms of Spearman $R^2$ at Basrah and %Bias at Mosul.

**Table 5.** Effectiveness of various satellite rainfall data in reproducing daily in situ rainfall in three cities in Iraq.

| Indices | City | GSMaP_GC | GSMaP_NRT | IMERG |
|---|---|---|---|---|
| Spearman $R^2$ | Baghdad | **0.339** | 0.259 | 0.292 |
| | Basrah | 0.505 | 0.519 | **0.521** |
| | Mosul | **0.624** | 0.593 | 0.539 |
| %Bias | Baghdad | **65.3** | 213.8 | 453.6 |
| | Basrah | 37.6 | −**16.1** | 196 |
| | Mosul | 27.7 | −29.4 | **18.5** |
| Skill Score | Baghdad | **0.278** | 0.159 | 0.137 |
| | Basrah | 0.419 | **0.545** | 0.278 |
| | Mosul | **0.466** | 0.325 | 0.371 |

Note: Bold numbers indicate the best performance.

Figure 4 shows boxplots of the in situ and satellite rainfall data at three locations. Generally, Mosul, located in the north, receives higher and more extreme rainfall than the

other two cities. None of the products could replicate the high rainfall events in Mosul. In contrast, GSMaP NRT and IMERG showed rainfall events at Baghdad of more than 200 mm, which generally does not occur in the cities. IMERG also showed very high rainfall events at Basrah, where rainfall is generally very low. Overall, the boxplot of GSMaP GC was closest to the boxplot of in situ rainfall.

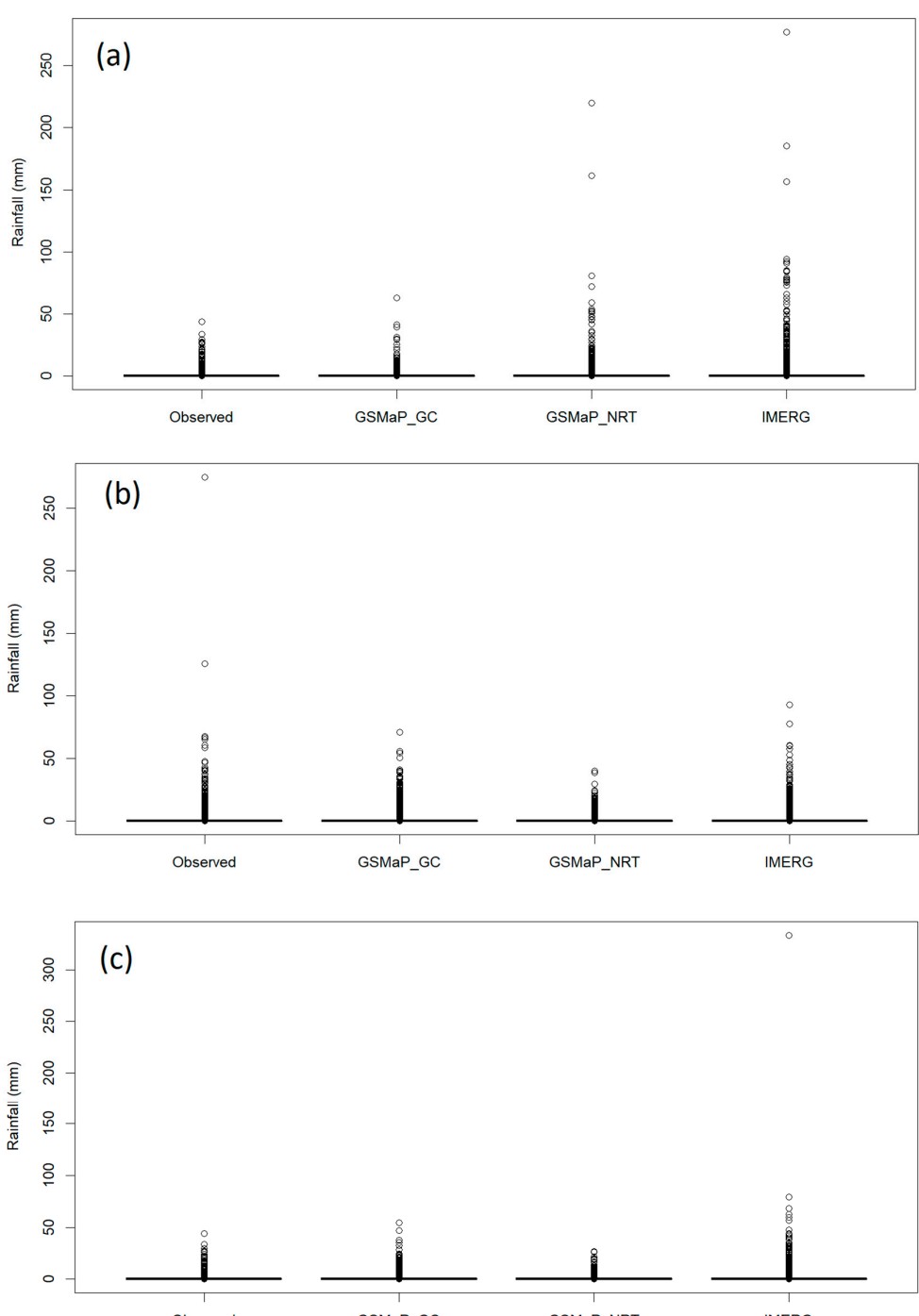

**Figure 4.** Box plot of in situ and satellite daily rainfall during 2000–2015 at (**a**) Baghdad, (**b**) Musol, (**c**) Basrah.

The Taylor diagram (shown in Figure 5) illustrates the effectiveness of satellite-based products. This diagram can summarize the performance of different satellite precipitation datasets based on the similarity between correlation and variability. The in situ rainfall is represented by a circle on the x-axis, while the satellite data are characterized by filling color circles. The optimal product is closest to the in situ data. Taylor's Diagram also showed that GSMaP GC is the best dataset for Iraq's major cities. It was very close to the observation at Baghdad. The performance of GSMaP GC and GSMaP NRT was similar at Basrah, but the variability of GSMaP GC was closer to the observation than that for GSMaP NRT.

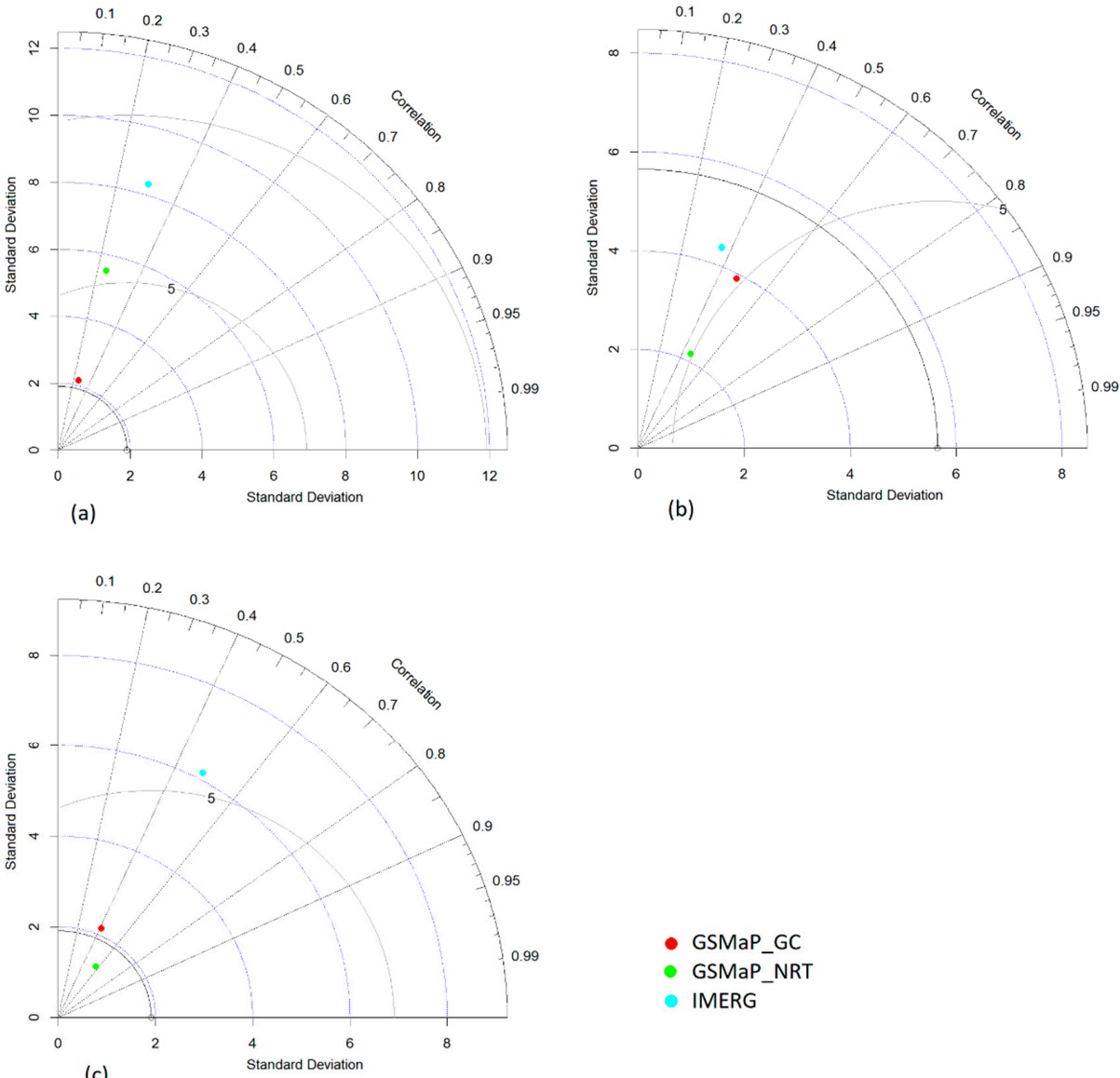

**Figure 5.** Taylor diagram of in situ and satellite daily rainfall during 2000–2015 at (**a**) Baghdad, (**b**) Musol, (**c**) Basrah.

The probability distribution functions (PDFs) of observed and satellite rainfall data are presented in Figure 6. Presenting a PDF of highly skewed data like daily rainfall is always difficult. Most days, the rainfall is zero, particularly in arid regions like Iraq, while high rainfall events are rare. This makes the PDF of the daily rainfall of Iraqi cities highly skewed, which is very difficult to judge graphically. Therefore, this study split the PDFs at each station into two parts, one representing low rainfall (≤1 mm) and the other the

rainfall more than 1 mm. This made it easy to visualize the similarity of the PDF of different satellite rainfall with the PDF of observed rainfall. The figure shows that the PDF of GSMaP GC is closer to the observed PDF at all three locations.

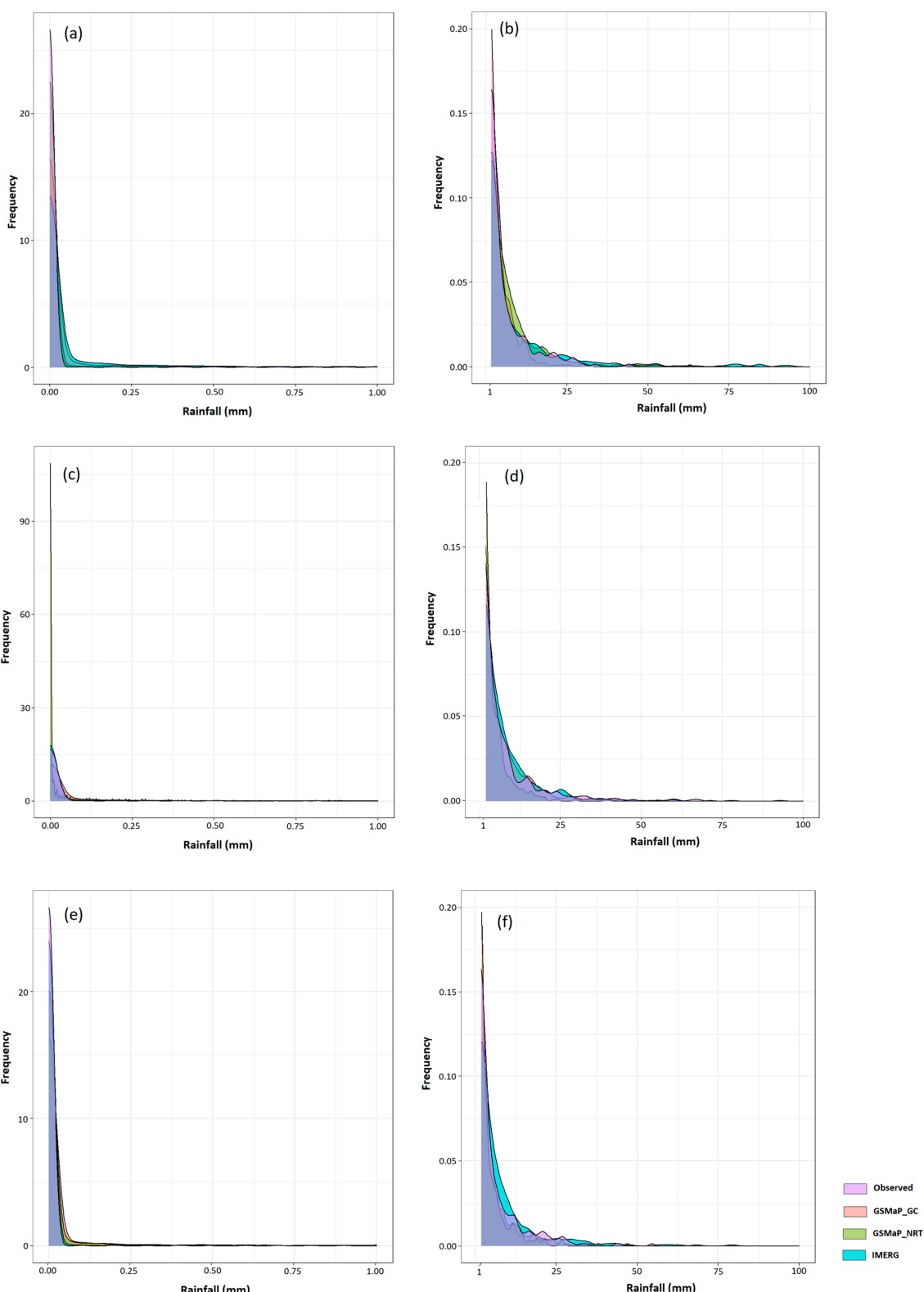

**Figure 6.** Distribution plots of in situ and satellite daily rainfall data at Baghdad (**a**,**b**), Mosul (**c**,**d**), and Basrah (**e**,**f**).

Finally, the satellite products were tested on how well they could predict the yearly rainfall maximums that had been seen. For this purpose, the one-, two-, and three-day cumulative rainfall maximum for all the years of the products were estimated and graphically presented in Figure 7. GSMaP GC was most accurate in capturing the rainfall maxima at Baghdad and Basrah. However, no product could capture the high rainfall events in Mosul.

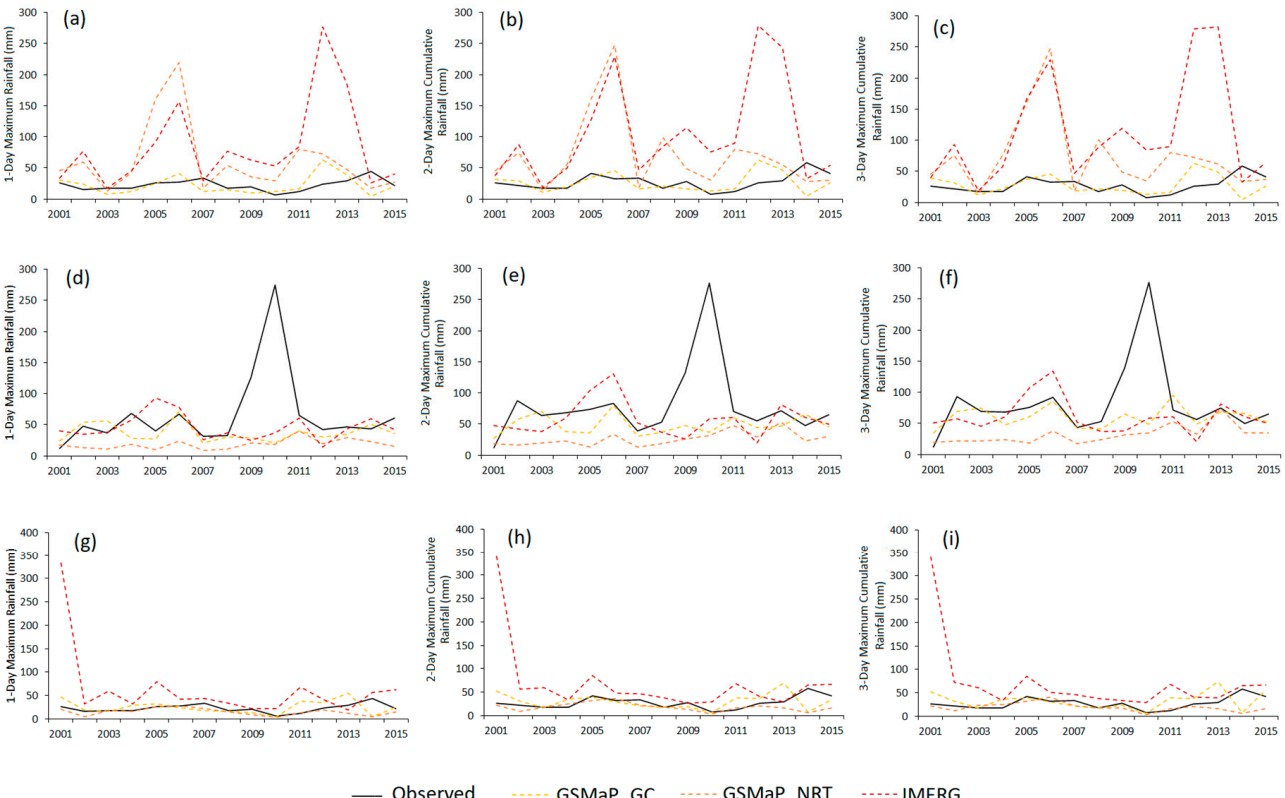

**Figure 7.** Annual rainfall maxima of in situ and satellite daily rainfall data for 1-, 2-, and 3-days at Baghdad (**a**–**c**), Mosul (**d**–**f**), and Basrah (**g**–**i**).

Overall, the analysis revealed GSMaP GC as the best product among the three satellite rainfall products considered in this study to capture the observed rainfall. Particularly, it performs very well in locations receiving low rainfall. However, it could not capture the high rainfall in the northern high-rainfall receiving station.

### 4.2. The Goodness of Fit Test

The annual maximum rainfall of GSMaP NRT, GSMaP GC, and IMERG at three stations for 1-, 2-, 3-, 4-, 6-, 12-, 24-, 72-, 96-, and 144-h durations were fitted with GP, GEV, and Gumbel. Table 6 illustrates negative log-likelihood results for Baghdad. The results showed that the GEV performs better for most rainfall durations at all stations. The performance of Gumbel was similar to GEV for rainfall durations 1, 2, 3, 4, 6, 12, and 24 h but higher for other periods at Baghdad. However, IMERG performed better than GP for 1-h rainfall durations. GSMaP GC performed similar to GEV for 144-h rainfall durations. The other two cities also showed a similar result to that obtained for Baghdad.

### 4.3. Generation of IDF Curves

Observed daily datasets and satellite-based precipitation datasets were used to generate IDF curves at all three major cities using the MLE-estimated GEV distribution parameters. Figure 8 illustrates the IDF curves produced using IMRG, GSMAP NRT, GSMAP GC, and the observed dataset. This study assessed the performance of satellite IDF curves based on the return durations of observed dataset 24-, 48-, and 72-h rainfall intensities. The results

showed that the IDF curves of all products were significantly different from the hourly IDF curve estimated using the observed dataset, except for GSMaP GC. In Baghdad, the observed 100year return rainfall for 24-, 48-, and 72-h duration was 3.8, 2.7, and 2.0 mm/h, respectively. The GSMaP GC estimated those 5.5, 2.7, and 1.6 mm/h. In contrast, GSMaP NRT estimations were 22.0, 10.0, and 6.5 mm/h. All products severely overestimated the IDF curve at Baghdad, except GSMaP GC, which slightly underestimated the IDF curves.

**Table 6.** Negative Log-likelihood values of different distributions at Baghdad for GSMaP NRT, GSMaP GC, and IMERG.

| Product | Distribution | Negative Log-Likelihood Statistics (MLE Estimator) Duration (h) | | | | | | | | | | |
|---------|--------------|------|------|------|------|------|------|------|------|------|------|------|
| | | 1 | 2 | 3 | 4 | 6 | 12 | 24 | 48 | 72 | 96 | 144 |
| GSMAP NRT | GEV | 93.1 | 101.5 | 106.7 | 110.5 | 113.4 | 117.7 | 120.6 | 123.0 | 123.1 | 123.5 | 123.8 |
| | Gumbel | 110.7 | 116.3 | 118.4 | 121.1 | 121.1 | 122.4 | 124.2 | 125.3 | 125.1 | 125.2 | 125.4 |
| | GP | 101.0 | 108.9 | 112.7 | 115.8 | 118.3 | 121.4 | 123.6 | 125.2 | 126.5 | 126.1 | 126.6 |
| GSMAP GC | GEV | 49.9 | 63.1 | 69.4 | 74.6 | 81.8 | 87.6 | 91.6 | 93.9 | 94.9 | 95.8 | 97.8 |
| | Gumbel | 50.7 | 64.8 | 72.2 | 77.7 | 83.7 | 88.4 | 92.7 | 94.2 | 95.1 | 95.9 | 97.8 |
| | GP | 54.4 | 69.4 | 76.2 | 81.1 | 86.7 | 92.7 | 96.6 | 97.6 | 97.3 | 96.9 | 96.4 |
| IMERG | GEV | 62.1 | 75.8 | 84.1 | 89.1 | 93.4 | 102.6 | 111.0 | 117.9 | 119.8 | 122.1 | 122.4 |
| | Gumbel | 62.1 | 75.9 | 84.2 | 89.1 | 93.5 | 103.1 | 112.9 | 120.2 | 121.6 | 123.9 | 123.9 |
| | GP | 53.6 | 79.2 | 88.3 | 89.1 | 101.4 | 114.0 | 122.8 | 127.2 | 127.9 | 129.1 | 129.4 |

The results were similar in the other two cities. GSMaP GC was slightly underestimated, but the other two products significantly underestimated the observed IDF curve at the Basrah station. GSMAP GC also slightly underestimated the IDF curve at the Mosul station, and the other products significantly overestimated it. Overall, the results showed that GSMaP GC showed the most realistic result through slightly underestimated rainfall intensities.

*4.4. IDF Curved Based on Sherman Equation*

The Sherman equation generated IDF curves for observed and satellite (GSMaP NRT, GSMaP GC, and IMERG) precipitation data from 2000 to 2022 for three major Iraqi cities are shown in Figure 9. In Baghdad, the observed 2-year return period observed rainfall based on the Sherman equation was 2.5, 1.8, 1.6, and 0.9 mm/h for 1-, 2-, 3-, and 6-h durations, respectively. The GSMaP GC computed those values as 2.6, 1.8, 1.5, and 0.1 mm/h; GSMaP NRT as 5.4, 3.2, 2.5, 1.3 mm/h; and IMERG as 7.4, 5.6, 4.7, and 3.4 mm/h, respectively. It indicates that GSMaP GC based on the Sherman equation provides the most realistic IDF curves. GSMaP GC also provided the best IDF compared to observation in Basrah and Mosul. There was a slight underestimation by GSMaP GC in all cities. However, it was negligible compared to the large overestimation by IMERG and GSMaP NRT.

The difference between the fitted IDF curves of GSMaP NRT, GSMaP GC, and IMERG with observation for major Iraqi cities are shown in Figures 10–12. The difference at Baghdad for GSMaP GC ranged from −100 to 5 mm/h. In contrast, it was between −10 and 170 mm/h for GSMaP NRT and −100 and 10 for IMERG. GSMaP GC and IMERG underestimated the observed IDF curves, while GSMaP NRT overestimated the IDF curves. However, the under- and overestimation were much less for GSMaP GC.

The results were similar at the other two locations. However, the differences were less compared to Baghdad. The difference for GSMaP GC ranged from −15 to 1 at Basrah and −70 to 10 a Mosul. The difference for GSMaP GC was much less than the other two products. This indicates the suitability of GSMaP GC IDF curve to be corrected for reliable estimation of IDF from satellite data.

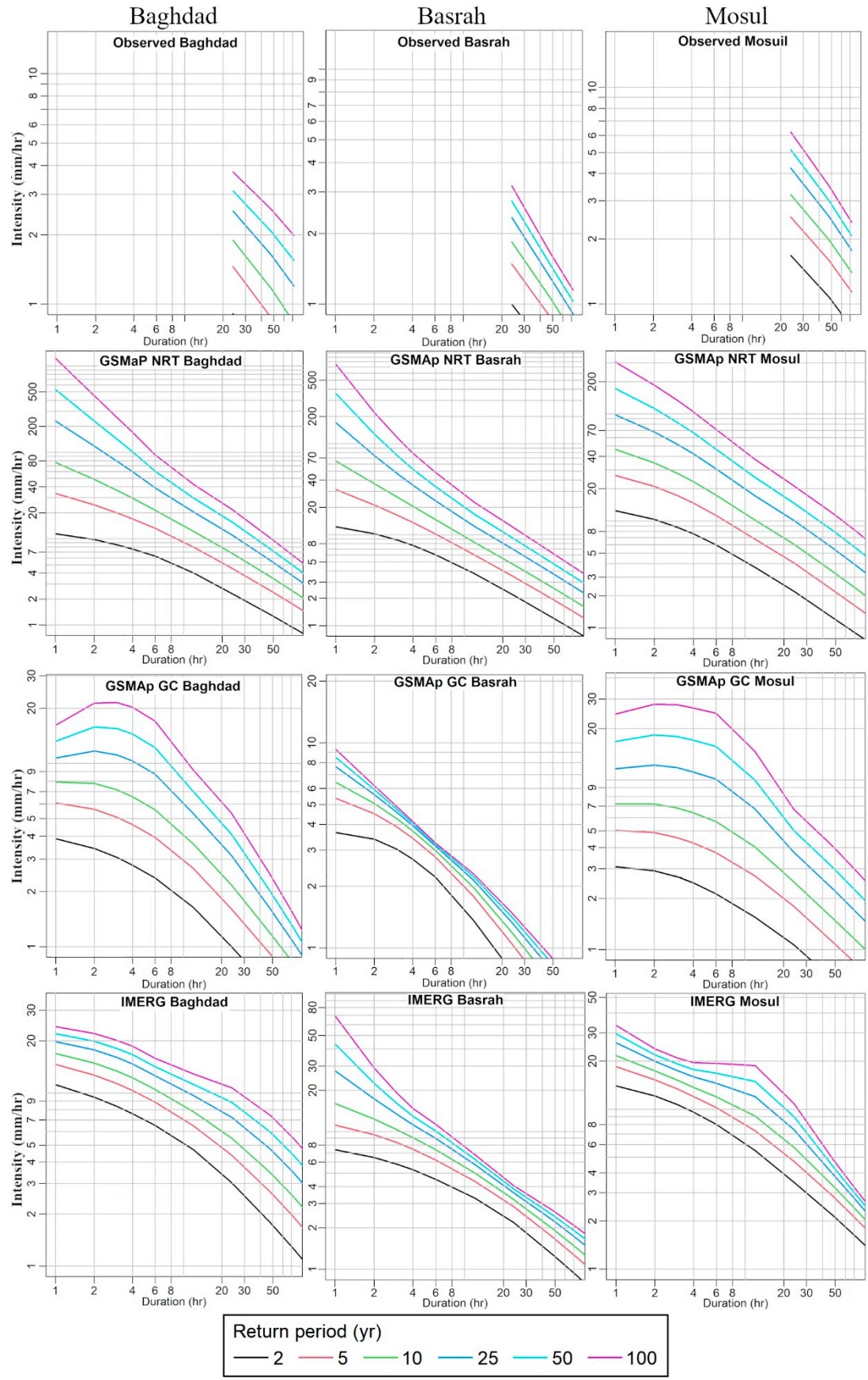

**Figure 8.** IDF curves at three Iraqi cities using satellite (GSMAp NRT, GSMAp GC, and IMERG) and observed rainfall data from 2000 to 2022.

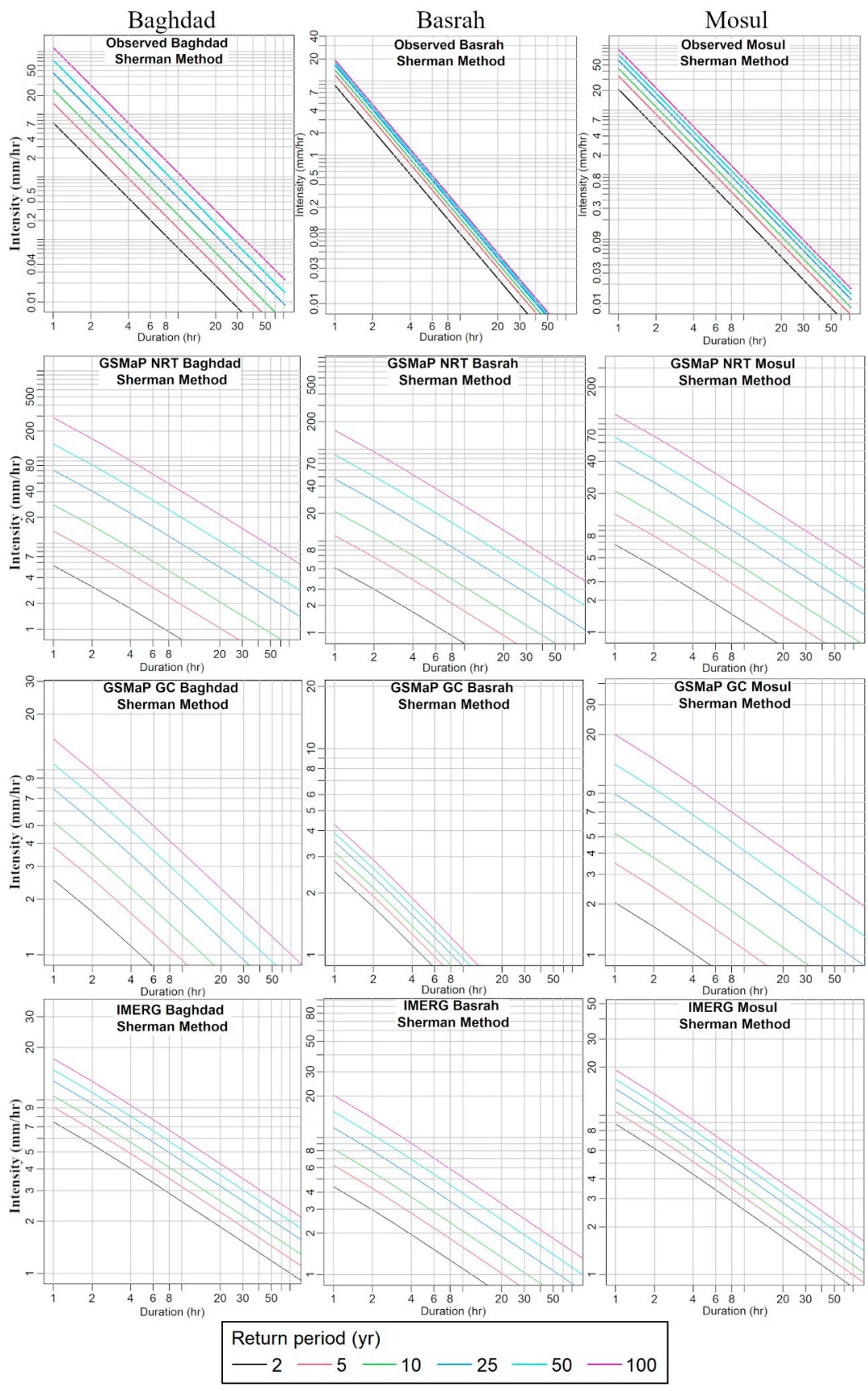

**Figure 9.** IDF curves at three Iraqi cities using satellite (GSMaP NRT, GSMaP GC, and IMERG) and observed rainfall data from 2000 to 2022 based on the Sherman method.

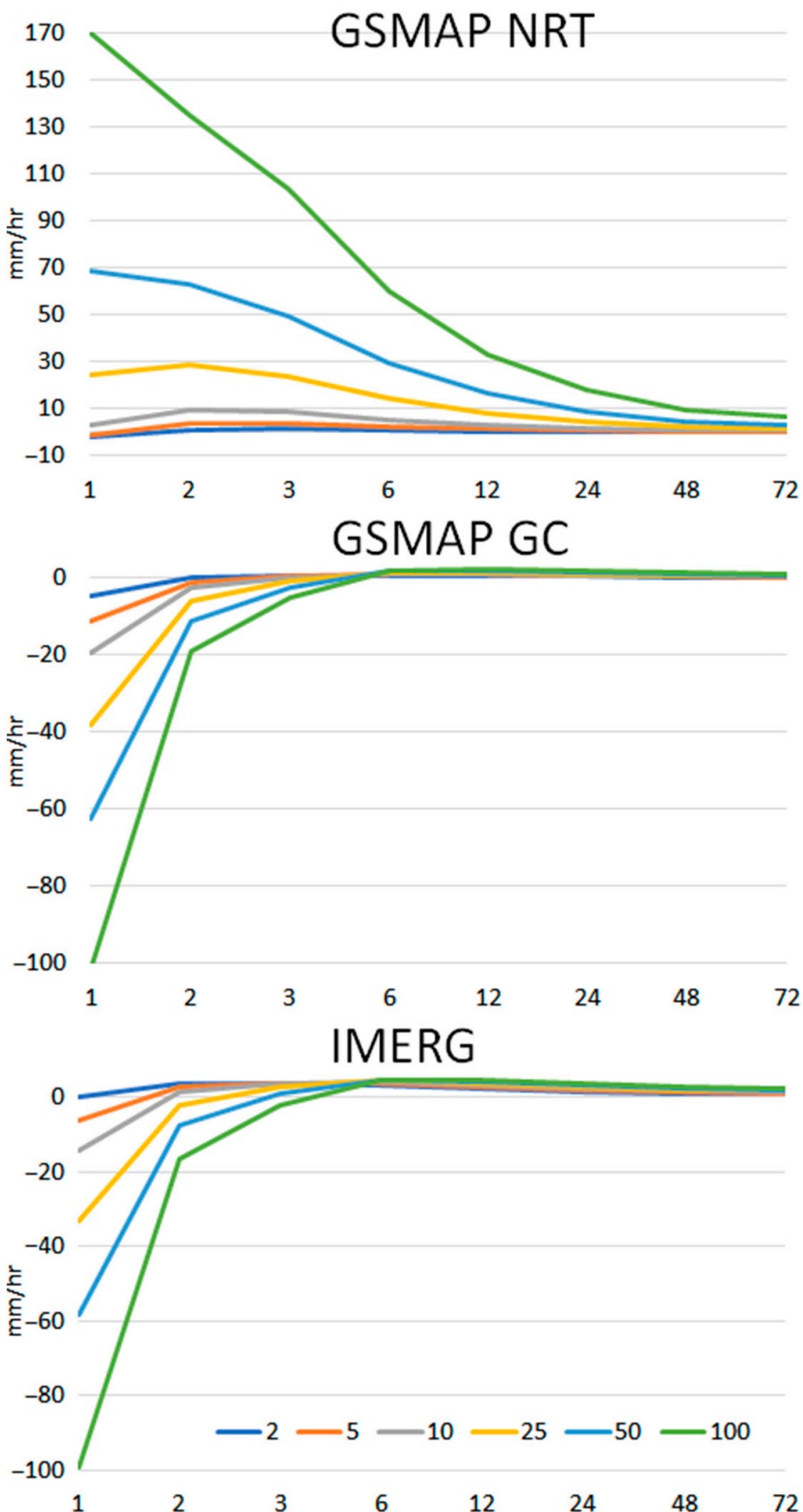

**Figure 10.** Bias in IDF curves in GSMaP NRT, GSMaP GC, and IMERG data compared to observation at Baghdad.

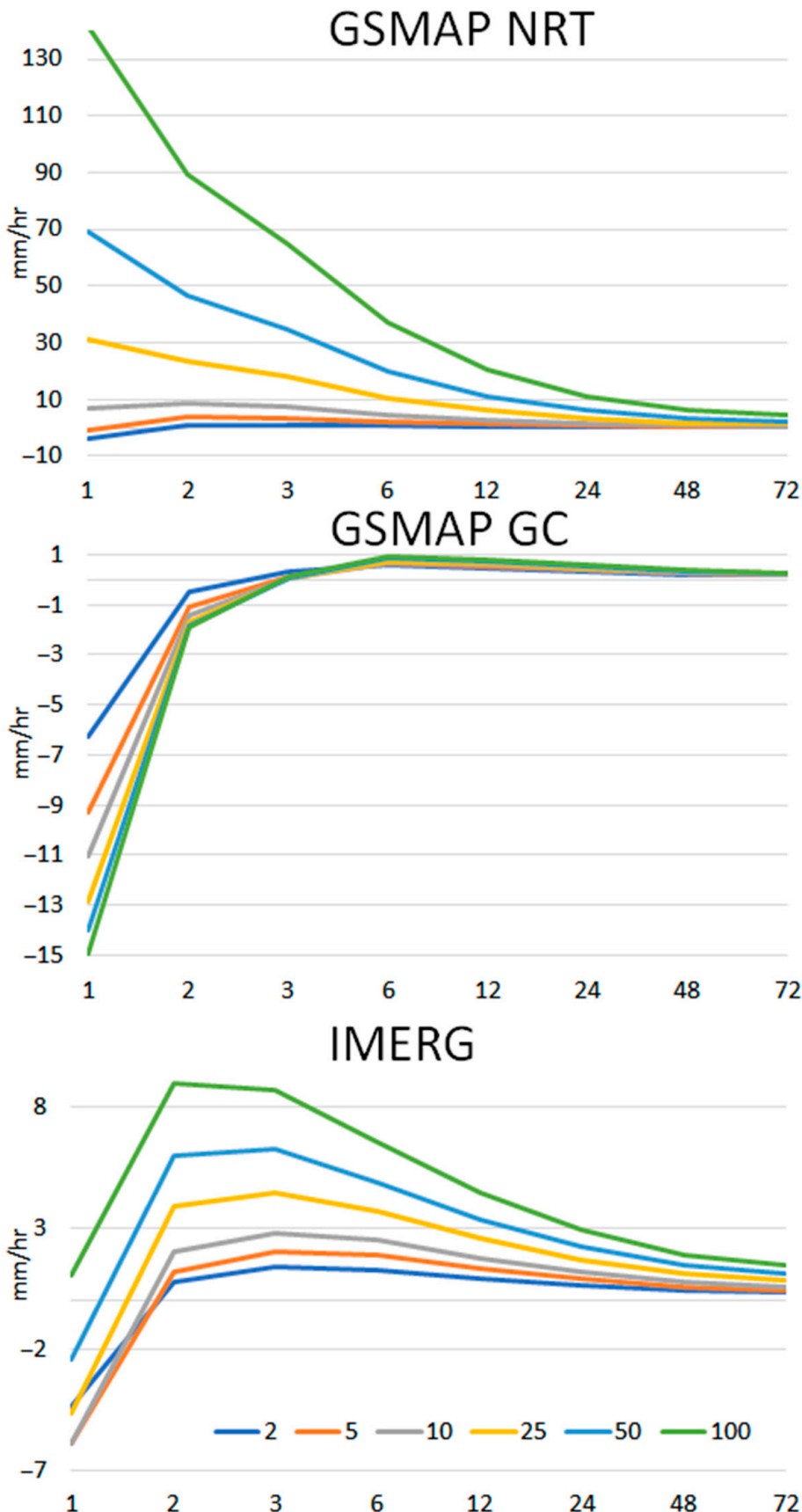

**Figure 11.** Bias in IDF curves in GSMaP NRT, GSMaP GC, and IMERG data compared to observation at Basrah.

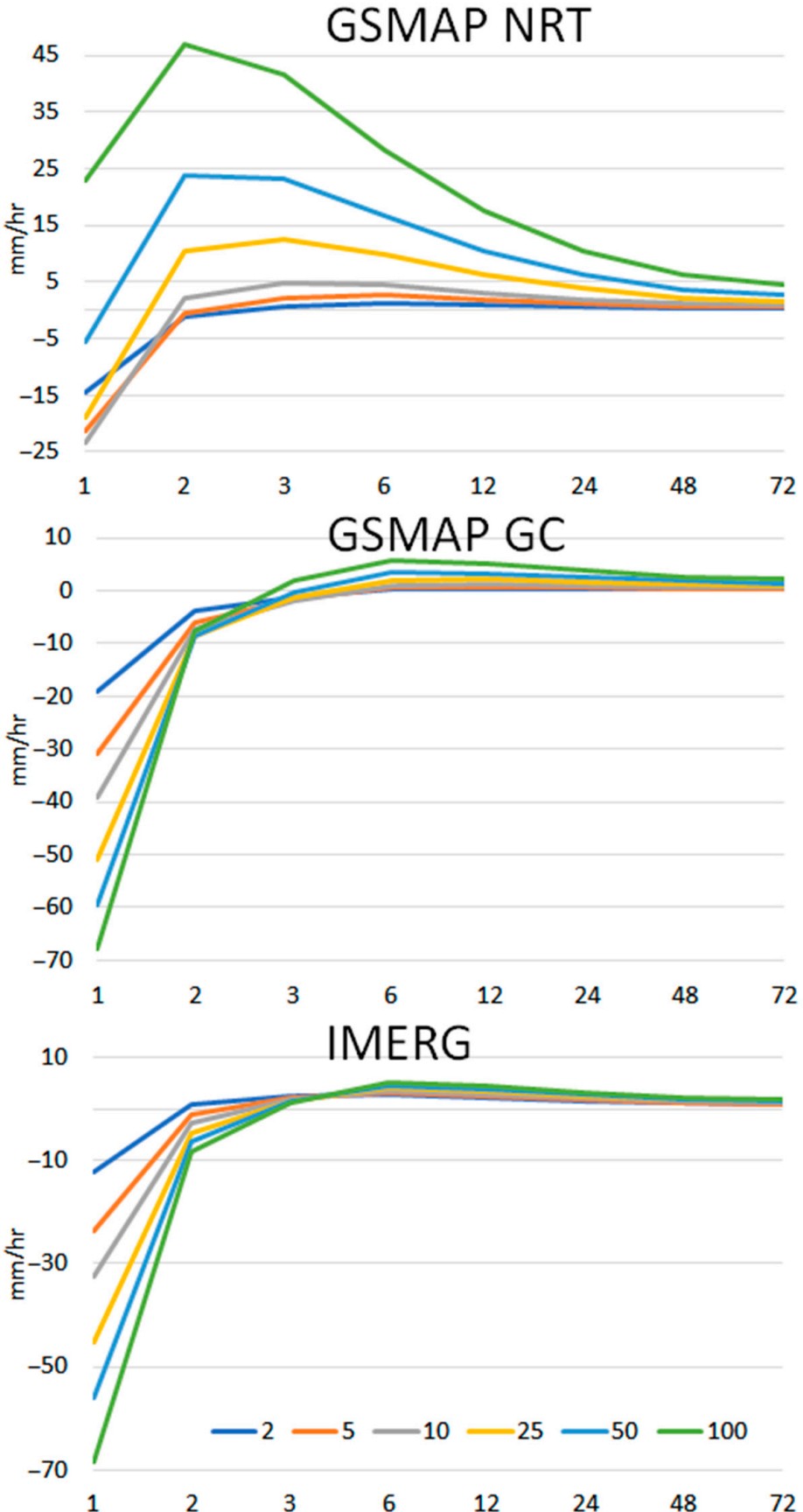

**Figure 12.** Bias in IDF curves in GSMaP NRT, GSMaP GC, and IMERG data compared to observation at Mosul.

## 5. Discussion

Both statistical analysis and visual assessment of satellite rainfall yielded the same results for the research. The results revealed GSMaP GC is far superior to other satellite precipitation products in Iraq. When comparing the three cities, the maximum GSMaP GC daily rainfall estimates were more in line with the observed maximum daily rainfall estimates. The distribution of outliers in GSMaP GC was comparable to that observed. The Taylor Diagram further demonstrated that GSMaP GC is the superior dataset for the major cities in Iraq. GSMaP GC showed higher correlations with observed data at Baghdad, Basra, and Mosul. IDF curves for GSMaP GC were less biased than IDF curves for other products. This demonstrates the viability of adjusting the GSMaP GC IDF curve for accurate calculation of IDF using satellite data.

No study has been conducted earlier to calculate the IDF using satellite rainfall data in Iraq. Kareem et al. [56] created the IDF curves for Erbil city using IDF curves and empirical IDF formulas using metrological stations. Mahdi et al. [57] compared different probability distributions to update the IDF curve of Baghdad city. They discovered that the heaviest rainfall ever recorded happened at a duration of 0.25 h with a return time of 100 years. However, several studies showed the ability of remote sensing data for hydrological studies in Iraq. Marra et al. [58] compared the radar with satellite data in nearby countries. They proved the ability of remote sensing datasets to provide quantitative information on previously unmapped regions of the planet. Suliman et al. [21] evaluated the skill of remote sensing rainfall products to characterize droughts in Iraq and showed higher performance of PERSIANN data to study droughts in the country. Jaber et al. [59] also showed the efficacy of remote sensing precipitation in rainfall-runoff modelling in Iraq. Studies in the nearby region also showed the applicability of satellite rainfall in hydrological applications in Jordan [60], Iran [61], Saudi Arabia [62,63], and Turkey [64].

Our analysis found that IMERG had exaggerated the rainfall intensity in Baghdad and Mosul. This is consistent with the findings of Chen et al. [65]. They looked at global satellite precipitation products and found that IMERG inflated its estimates of heavy rain. In contrast to other data, the IMERG regression line was near the diagonal. This is because IMERG provided more accurate estimates of heavy precipitation than competing data sources. IMERG showed more bias compared to the observed in the current study. However, IMERG precipitation biases are larger since they incorrectly predict less-than-average downpours.

Satellite rainfall data in Iraq have previously been evaluated [21,23,66,67]. However, there has been no comprehensive analysis of satellite rainfall systems like IMERG and GSMaP. As a result, it was not possible to draw any comparisons between our study and other studies. However, studies in nearby regions also showed the higher capability of GSMaP in replicating in situ rainfall. Saber and Yilmaz [64] assessed GSMaP rainfall with in situ data to understand their skill in modeling seasonal, annual, and spatial rainfall distribution. They found a good linear association of GSMaP rainfall with observation. Darand and Fathi [61] showed the higher potential of GSMaP in characterizing droughts in Iran. Ghorbanian et al. [68] evaluated the skill of six satellite products in Iran for the last 20-year period and showed better performance of GSMaP precipitation than other datasets.

Remote sensing rainfall data can be biased in several ways, depending on the specifics of the underlying physiography. This includes the terrain, height, proximity to the shorelines, and climatic aspects like the speed of the wind and the sort of clouds present [28]. Future research should focus on linking specific physiographic and meteorological characteristics with the identified bias in remote sensed rainfall to better understand the numerous variables affecting the bias. After these variables are considered, IDF curves derived from remote sensing precipitation products can be more accurately estimated thanks to a bias correction procedure [28].

## 6. Conclusions

This study aimed to create IDF curves for three major cities in Iraq from remote sensing rainfall data. The lack of current rainfall-based updated IDF curves makes it difficult to update SWMS and reduce rising urban floods in Iraq. Sustainable urban SWM in the country may be a major step closer with the help of the developed IDF curves. The intention was to select the best satellite rainfall product which could be used for developing IDF curves for major cities of Iraq so that they could be updated from time to time without depending on in situ observation. This study used observed daily data to evaluate the performance of different satellite rainfall products considering the unavailability of hourly rainfall data in Iraq. Different PDFs were fitted to ARI time series three satellite products to identify the best PDF, which was subsequently employed to develop IDF curves at different cities. The study revealed GEV as the best PDF for estimating hourly ARI distribution parameters at all studied locations in Iraq. GSMaP GC is the best product to replicate observed daily rainfall. GSMaP GC showed higher correlations with observed data at Baghdad, Basra, and Mosul. The under- and overestimation were much less for GSMaP GC. It is also the most reliable in estimating IDF curves in different cities. The IDF curves generated using GSMaP GC were more realistic than that produced by the other two products. Therefore, this study suggests the correction of GSMaP GC IDF biases based on the percentage of difference estimated in this study to generate IDF curves for major cities of Iraq. The study also indicates that bias-corrected IDF curves can be used to estimate the recurrence of different intensities of rainfall events at locations where in situ data are unavailable. It can help design hydraulic structures for mitigating the growing impacts of climatic extremes like floods in Iraq due to climate change. The IDF curves were established for only three of Iraq's most important cities in this study. This process, however, can be performed in any other satellite rainfall grid position within the city to generate an IDF curve for that area. The procedure developed in this study can be extended to generate IDF curves at ungauged areas of Iraq using satellite rainfall. This is important considering the rapid urbanization but the unavailability of hourly or sub-hourly in situ rainfall data in the country. In the future, other satellite rainfall data can be considered to find the best product for developing IDF curves for Iraqi cities. In addition, global climate model simulations can be used for the projections of IDF under the future climate of Iraq.

**Author Contributions:** Conceptualization, S.J.Z., X.W. and S.S.; methodology, S.J.Z., M.M.H. and S.S.; software, M.M.H. and S.S.; validation, M.M.H., X.W. and S.S.; formal analysis, S.J.Z. and S.S.; investigation, S.J.Z., M.M.H. and S.S.; resources, S.J.Z. and S.S.; data curation, S.J.Z. and M.M.H.; writing—original draft preparation, S.J.Z., M.M.H., X.W. and S.S.; writing—review and editing, S.J.Z., M.M.H., X.W. and S.S.; visualization, M.M.H. and S.S.; supervision, X.W. and S.S.; project administration, X.W. and S.S.; funding acquisition, X.W. and S.S. All authors have read and agreed to the published version of the manuscript.

**Funding:** The Belt and Road Special Foundation of the State Key Laboratory of Hydrology-Water Resources and Hydraulic Engineering (Nos. 2019491311 and 2020491011) and Young Top-Notch Talent Support Program of National High-level Talents Special Support Plan.

**Data Availability Statement:** All data generated or used during the study are available from the corresponding author by request.

**Acknowledgments:** Authors are grateful to the Japan Science and Technology Agency (JSTA) for the free access to satellite precipitation products through the web portal.

**Conflicts of Interest:** The authors declare no conflict of interest.

## Abbreviations

| Symbol | Definition |
| --- | --- |
| IDF | Intensity-duration-frequency |
| PDFs | Probability distribution functions |
| SWMS | Stormwater management systems |
| MENA | Middle East and North Africa |

| IMERG | Integrated Multi-Satellite Retrievals for Global Precipitation Measurement |
| GSMaP | Global Satellite Mapping of Precipitation |
| GSMaP NRT | Global Satellite Mapping of Precipitation Near Real-Time |
| GSMaP GC | Global Satellite Mapping of Precipitation gauge corrected |
| GPM | Global Precipitation Measurement |
| CREST | Core Research for Evolutional Science and Technology |
| JSTA | Japan Science and Technology Agency |
| JAXA | Japan Aerospace Exploration Agency |
| PMM | Precipitation Measuring Mission |
| ARI | Annual rainfall intensity |
| GEV | Generalized Extreme Value |
| GP | Gumbel and General Pareto |
| MLE | Maximum Likelihood |
| $R^2$ | Spearman coefficient of determination |
| %BIAS | percentage of bias |
| SS | Perkin's skill score |

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
