# Peer review of "Utilizing Satellite Data to Establish Rainfall Intensity-Duration-Frequency Curves for Major Cities in Iraq"

_water, doi:10.3390/w15050852_

Round 1
Reviewer 1 Report
The manuscript titled Utilizing Satellite Data to Establish Rainfall Intensity-Duration-Frequency Curves for Major Cities in Iraq after Major Revision.
In my opinion, the topic of this paper is relevant to the Journal Water MDPI after major revisions and approved moderate corrections.
The topic of the paper is very interesting and important, especially in the context of remote sensing in analysis and long-term precipitation characteristics.
The journal Water MDPI wants interesting and high quality papers.
First, the paper has the next sections and subsections (i.e., abstract, introduction, study area and data, Iraq region, data set, methodology, research steps, distribution functions, Sherman equation, evaluation of satellite precipitation data, results, performance of satellite precipitation data, goodness-of-fit test, generation of IDF curves, IDF curves based on Sherman equation, conclusion, etc.)
I strongly recommend that the authors include a section on abbreviations because of the large number of short words (scientific terms).
The section of Abstract
The summary section does not fully explain the main findings of this research. The authors need to add the main findings and a sentence to explain how this research impacts issues such as satellite data, rainfall intensity, duration, and frequency, especially in the area of Iraq. It is not usual to use short terms (abbreviations) in the summary. It is better to use original terms and then put them in the introduction section (abbreviations).
I highly recommend the author to use long terms in the section and then short terms (abbreviations) in the introduction section.
The section of Introduction
In the first part of this section, it would be good to cite (explain) IPCC - Intergovernmental Panel on Climate Change. This website has a lot of data tracking precipitation events. Please check them all.
Line 42, the sentence intensity-duration-frequency, can the authors better explain the duration of frequency?
In the section of introduction, the author must provide few sentences which explain ho
Can the authors in this section place comparison between Sentinel-1 or Sentinel-2 with this satellite methodology?
Line 72, is this a more accurate resolution of the data or not?
It is not recommended to put the table in the introduction section. Please move table 1 to another place.
In the Introduction section, I strongly recommend that the authors add two valuable references to support this research.
Two references are
. - Ombadi, M., Nguyen, P., Sorooshian, S., & Hsu, K. (2018). Developing intensity-duration-frequency (IDF) curves from satellite-based precipitation: Methodology and evaluation. Water Resources Research, 54, 7752– 7766. https://doi.org/10.1029/2018WR022929.
- Valjarević, A.; Morar, C.; Živković, J.; Niemets, L.; Kićović, D.; Golijanin, J.; Gocić, M.; Bursać, N.M.; Stričević, L.; Žiberna, I.; Bačević, N.; Milevski, I.; Durlević, U.; Lukić, T. Long Term Monitoring and Connection between Topography and Cloud Cover Distribution in Serbia. Atmosphere 2021, 12, 964. https://doi.org/10.3390/atmos12080964.
The section of Study area and data
Iraq region
Perhaps it is better to change this title to Physical and Climatological Properties of Iraq?
Figure 1,
Excellent illustration, but if this illustration belongs to the authors, please write a few sentences how this illustration was created.
Figure 2: This figure was used to try to show the amount of precipitation in Iraq. I think the pixels are too big, can the pixels be made smaller?
Line 125, in which resolution the data was downloaded.
In line 129, the authors state that they use Google Earth Engine. Can the authors explain more about this process?
The resolution of figure 3 is not good enough, can the authors fix it?
Line 158, Why did the authors Gumbel and General use Pareto and how can this analysis be compared to the exponential distribution?
The section of Evaluation of satellite precipitation data
In this section it will be good to explain the authors (compared) all methods with precipitation retrieval algorithms.
The section of Generation of IDF curves
How the IDF curves projected future climate of Iraq?
The section of Discussion
This section must be extended.
The section of the conclusion
In this section, the authors must answer the following questions?
Why is this research important?
Did the authors find a relationship between the satellite data analyzed and precipitation intensity, duration, and frequency?
Please include other results and main objectives in the conclusion section.
This work has the potential to be published. The authors have done a lot in this manuscript. The work is very interesting and scientifically correct.
In the end, I recommend a major revision.
Good luck to the authors
The reviewer#2

Author Response
The manuscript titled Utilizing Satellite Data to Establish Rainfall Intensity-Duration-Frequency Curves for Major Cities in Iraq after Major Revision.
In my opinion, the topic of this paper is relevant to the Journal Water MDPI after major revisions and approved moderate corrections. The topic of the paper is very interesting and important, especially in the context of remote sensing in analysis and long-term precipitation characteristics. The journal Water MDPI wants interesting and high quality papers.
Answer: Thank you very much for your comments. We revised the paper based on your comments. It helped us to improve the quality of the manuscript significantly. All of your comments have been incorporated in the revised manuscript. Details of the revisions are mentioned in point-to-point answers to your comments below. Besides, all the changes in the manuscript are marked with red-coloured text.
- First, the paper has the next sections and subsections (i.e., abstract, Introduction, study area and data, Iraq region, data set, methodology, research steps, distribution functions, Sherman equation, evaluation of satellite precipitation data, results, performance of satellite precipitation data, goodness-of-fit test, generation of IDF curves, IDF curves based on Sherman equation, conclusion, etc.)
I strongly recommend that the authors include a section on abbreviations because of the large number of short words (scientific terms).
Answer: Thank you for your comment. We added a list of abbreviations at the end of the revised manuscript.
- The summary section does not fully explain the main findings of this research. The authors need to add the main findings and a sentence to explain how this research impacts issues such as satellite data, rainfall intensity, duration, and frequency, especially in the area of Iraq. It is not usual to use short terms (abbreviations) in the summary. It is better to use original terms and then put them in the introduction section (abbreviations).
I highly recommend the author to use long terms in the section and then short terms (abbreviations) in the introduction section.
Answer: We modified the abstract to use the long terms when they are used for the first time. We also added more results to the abstract. The main findings of the comparison of satellite data and their ability to estimate rainfall in a predominantly arid region like Iraq are also provided based on your suggestions. The revised abstract is as below:
“This study generates intensity-duration-frequency curves for three important cities in Iraq using Global Precipitation Measurement Integrated Multi-Satellite Retrievals for Global Precipitation Measurement (IMERG), Global Satellite Mapping of Precipitation near real-time (GSMaP NRT) and gauge corrected (GSMaP GC) satellite precipitation datasets. Many probability distribution functions were used to fit the maximum yearly rainfall data. The Sherman equation was used to create intensity-duration-frequency (IDF) curves for rainfall intensities with 2, 5, 10, 25, 50, and 100-year return periods, with the estimated coefficients of the best-fit distribution serving as the fitting parameters. The discrepancy between the IDF curves produced from the satellites and the observed data to bias correct the satellite IDF curves. The Generalized Extreme Value Distribution model best describes the hourly rainfall distribution of satellite data. GSMaP GC was the best option for creating IDF curves with higher correlations with observed data at Baghdad, Basra, and Mosul. The study indicates the necessity of gauge correction of satellite rainfall data to reduce under- and over-estimating observed rainfall. GSMaP GC can reasonably estimate rainfall in a predominantly arid climate region like Iraq. The generated IDF curves may be an important step toward achieving sustainable urban stormwater management in the country.”
- In the first part of this section, it would be good to cite (explain) IPCC - Intergovernmental Panel on Climate Change. This website has a lot of data tracking precipitation events. Please check them all.
Answer: Thank you for your recommendation. We cited IPCC report in the revised manuscript.
- Line 42, the sentence intensity-duration-frequency, can the authors better explain the duration of frequency?
Answer: Thank you very much for your comment. We explained it in the revised manuscript by adding the following texts:
“Such curves are developed from the time series of annual peak rainfall events for different durations. It allows for estimating the recurrence of different intensities of rainfall, which is required for estimating peak runoff for designing hydraulic structures (De Paola et al., 2014; Gámez-Balmaceda et al., 2020).”
- Can the authors in this section place comparison between Sentinel-1 or Sentinel-2 with this satellite methodology?
Answer: Sentinel used multispectral sensors. It is also not used for retrieval of precipitation data. Therefore, we not copared Sentinel with satellite rainfall retrieval methodology.
- Line 72, is this a more accurate resolution of the data or not?
Answer: We mentioned the absence of hourly and sub-hourly observed data, which made the hydrological researchers turn to satellite precipitation data in recent years.The revised sentence is as below:
“Without a dense rainfall monitoring network or better temporal resolution (hourly or sub-hourly data), hydrological researchers have increasingly turned to satellite precipitation data in recent years.”
- It is not recommended to put the table in the introduction section. Please move table 1 to another place.
Answer: Thanks for your comment. We moved table 1 to the study area section.
- In the Introduction section, I strongly recommend that the authors add two valuable references to support this research.
- Ombadi, M., Nguyen, P., Sorooshian, S., & Hsu, K. (2018). Developing intensity-duration-frequency (IDF) curves from satellite-based precipitation: Methodology and evaluation. Water Resources Research, 54, 7752– 7766. https://doi.org/10.1029/2018WR022929.
- Valjarević, A.; Morar, C.; Živković, J.; Niemets, L.; Kićović, D.; Golijanin, J.; Gocić, M.; Bursać, N.M.; Stričević, L.; Žiberna, I.; Bačević, N.; Milevski, I.; Durlević, U.; Lukić, T. Long Term Monitoring and Connection between Topography and Cloud Cover Distribution in Serbia. Atmosphere 2021, 12, 964. https://doi.org/10.3390/atmos12080964.
Answer: We added the mentioned references.
- Perhaps it is better to change this title to Physical and Climatological Properties of Iraq?
Answer: We changed it based on your comment.
- Figure 1, Excellent illustration, but if this illustration belongs to the authors, please write a few sentences how this illustration was created.
Answer: We used QGIS software to create the figure and we added a digital elevation model of Iraq to simulate the elevations of the study area. We mentioned it in the revised manuscript as below:
“The site of the cities on the map of Iraq is displayed in Error! Reference source not found., which was created using QGIS open source software. Iraq’s boundary and the digital elevation model (DEM) data were downloaded from the publicly accessible DIVA_GIS website (https://www.diva-gis.org).”
- Figure 2: This figure was used to try to show the amount of precipitation in Iraq. I think the pixels are too big, can the pixels be made smaller?
Answer: We changed the pixel's resolution to 0.1 degree to reduce the pixel size. Thank you for your suggestion. The figure is now looking better.
- Line 125, in which resolution the data was downloaded.
Answer: We estimated the IDF curves for three major cities. Therefore, grid points representing the cities of Iraq were used. We mentioned it clearly in the revised manuscript using following statements:
“This study used Google Earth Engine (GEE), a cloud-based system for universe geospatial research, to retrieve satellite precipitation data [39]. This platform provides satellite precipitation products and the required tools for downloading them for a specific area or point. This study downloaded the satellite rainfall datasets of the grid locations representing the cities of Iraq using GEE.”
- In line 129, the authors state that they use Google Earth Engine. Can the authors explain more about this process?
Answer: We used Google Earth Engine to download the hourly data of the three satellite products in each city under study. We elaborated it as below:
“This study used Google Earth Engine (GEE), a cloud-based system for universe geospatial research, to retrieve satellite precipitation data [39]. This platform provides satellite precipitation products and the required tools for downloading them for a specific area or point. This study downloaded the satellite rainfall datasets of the grid locations representing the cities of Iraq using GEE.”
- The resolution of figure 3 is not good enough, can the authors fix it?
Answer: We modified the figure with a 300-dpi resolution.
- Line 158, Why did the authors Gumbel and General use Pareto and how can this analysis be compared to the exponential distribution?
Answer: Thank you for your comment. There are many probability distribution functions (PDFs). In this study, we tried to consider the PDFs which are more appropriate for the study area. This was identified based on a literature review. This is mentioned in the article as follows:
“According to earlier studies in Iraq, one of these three PDFs typically offered the best fit for the ARI time series. For instance, Majeed et al. [20] demonstrated that the Gumbel provided the best rainfall intensities in Najaf city, Iraq, for various return periods and durations. AL-Dulaimi et al. [41] claimed that Gumbel distribution was the best frequency analysis technique in Babylon City And Alluvial Fertile Zone, Iraq. Different stations in northern Iraq showed different best PDFs, including GEV, Gumble and GP [42]. Thus, only these three PDFs were considered in the current investigation.”
- In this section it will be good to explain the authors (compared) all methods with precipitation retrieval algorithms.
Answer: Thank you for your comment. We discussed in the data description section (Section 2.2) using following texts:
“The precipitation data in all the satellite products used in this study was retrieved using a combination of multiple passive microwave and infra-red sensors [29]. IMERG has three rainfall products, early, late and final run. The IMERG final run (FR) is the most accurate of the three precipitation modes [32]. GSMaP NRT and GSMaP GC rainfall data are collected and compiled by the Core Research for Evolutional Science and Technology (CREST) of the Japan Science and Technology Agency (JSTA) in collaboration with the Japan Aerospace Exploration Agency (JAXA) Precipitation Measuring Mission (PMM) Science Team [33–35]. The former was developed by fusing cloud movement vectors derived from infrared photos with global precipitation rates derived from passive microwave radiometers [36,37]. The latter is a by-product of GSMaP NRT, developed by correcting it with precipitation of Climate Prediction Center [38].”
- The section of Generation of IDF curves. How the IDF curves projected future climate of Iraq?
Answer: The scope of the study was limited to the generation of IDF curves based on satellite data. We have not attempted to project IDF curves under the future climate of Iraq. However, such a study can be conducted in the future. We mentioned this as a recommendation at the end of the conclusion section, as below:
“Besides, global climate model simulations can be used for the projections of IDF under the future climate of Iraq.”
- The section of Discussion. This section must be extended.
Answer: We modified the section and added more texts in the discussion section.
- The section of the conclusion. In this section, the authors must answer the following questions?
Why is this research important?
Answer: Thanks for your comment. We added the importance of this study in the conclusion section. Please see the red-colored texts.
- Did the authors find a relationship between the satellite data analyzed and precipitation intensity, duration, and frequency?
Answer: The present study estimated IDF curves based on satellite rainfall considered in this study and observed rainfall. The results are shown in Figure 9. It presents the IDF curves at 3 Iraqi cities using satellite (GSMaP NRT, GSMaP GC and IMERG) and observed rainfall data from 2000 to 2022 based on the Sherman method. The IDF curves of the satellite rainfall and observed rainfall show the relationship between the satellite and observed rainfall intensity, duration and frequency. This is discussed in section 4.4 of the manuscript.
- Please include other results and main objectives in the conclusion section.
Answer: Thanks for your comment. The main objective is provided in the first line of the conclusion section. We added more results in the conclusion section.

Reviewer 2 Report
Dear Editor.
I have finished my review on the proposed paper “Utilizing Satellite Data to Establish Rainfall Intensity-Duration-Frequency Curves for Major Cities in Iraq”, water-2212494-peer-review-v1.
Summary of the manuscript:
In the proposed paper, the author’s goal is to examine the applicability of GPM IMERG, GSMaP NRT, and GSMaP GC satellite precipitation datasets, to generate intensity-duration-frequency (IDF) curves for three major cities of Iraq (Baghdad, Basrah and Mosul). GSMaP GC was the best option for creating IDF curves. However, most of these satellite products tend to underestimate the real intensity of rainfall.
General review:
1. Generally, the manuscript presents an interesting topic and the specific research seems to include some significant points for the research community of this field.
2. The proposed paper is very well written with very good use of English language. Except some very minor grammatical mistakes and word errors. The author should check again the paper to correct these minor mistakes.
3. The proposed paper is very well structured. It begins with the Introduction with some references that helps the reader to get into the subject immediately. In Introduction there is an effort to provide previous studies with similar scientific content, which took place in the research area and in other countries. Author describes and set very well the scientific problem and how other researchers have approached. At the end of Introduction, authors clearly state the goals of the research. However, I believe that for the specific subject you can enhance the provided literature (see below comments).
4. The methodology is generally very interesting, and well explained.
5. The results and the discussion are generally OK. However, there some parts that need revisions (see below comments).
Additional points for revision:
In my opinion, the proposed paper could be characterized as a very good research work, complies with aims of WATER.
79-80: Please, add here more literature.
Line 120: “IMERGE”? You mean IMERG?
Figure 3: In the flow chart you show some statistics that were not used in the study, for example the RMSE and the KGE. Or you should add in the statistical analysis these two statistics or correct the figure.
3.2 Distribution functions: I do not understand why you limit so much the distributions of the analysis (GEV, Gumbel, GP)? There are other distributions that may fit better to your data. You rely on other studies, but I think that this is not appropriate.
For the selected distributions: It is well known that the length of the rainfall time series in very significant for the selection of the appropriate distribution (doi.org/10.1007/978-3-319-35095-0_48 and doi.org/10.1023/A:1008001312219). Your time series are very short (20 years) and this may have an impact on the selection of the best distribution. For that reason, you have to test other distributions like the EV-1. Add in the text (lines 155-170) an explanation using the above proposed literature.
Lines 323-329: This paragraph is a repetition of the Introduction. I think that is not necessary and you can remove it.
Author Response
I have finished my review on the proposed paper “Utilizing Satellite Data to Establish Rainfall Intensity-Duration-Frequency Curves for Major Cities in Iraq”, water-2212494-peer-review-v1.
In the proposed paper, the author’s goal is to examine the applicability of GPM IMERG, GSMaP NRT, and GSMaP GC satellite precipitation datasets, to generate intensity-duration-frequency (IDF) curves for three major cities of Iraq (Baghdad, Basrah and Mosul). GSMaP GC was the best option for creating IDF curves. However, most of these satellite products tend to underestimate the real intensity of rainfall.
Generally, the manuscript presents an interesting topic and the specific research seems to include some significant points for the research community of this field.
Answer: Thank you very much for your encouraging comments. We have revised the paper based on your comments. All of your comments are considered in the revised manuscript. Details of the revisions made are mentioned below in point-to-point answers to your comments.
- The proposed paper is very well written with very good use of English language. Except some very minor grammatical mistakes and word errors. The author should check again the paper to correct these minor mistakes.
Answer: The language of the article is thoroughly revised. Any spelling and grammatical mistakes have been corrected. I hope you will find the language of the article sufficiently standard for publication in Water journal.
- The proposed paper is very well structured. It begins with the Introduction with some references that helps the reader to get into the subject immediately. In Introduction there is an effort to provide previous studies with similar scientific content, which took place in the research area and in other countries. Author describes and set very well the scientific problem and how other researchers have approached. At the end of Introduction, authors clearly state the goals of the research. However, I believe that for the specific subject you can enhance the provided literature (see below comments).
Answer: Thank you very much for your comment on the Introduction. We are happy to know that you found the Introduction sound. We revised the Introduction based on your comments.
- The methodology is generally very interesting, and well explained.
Answer: Thank you very much for your comment on the methodology
- The results and the discussion are generally OK. However, there some parts that need revisions (see below comments).
Answer: Thank you very much for your comment on the results and the discussion sections. We revised them based on your comments.
- In my opinion, the proposed paper could be characterized as a very good research work, complies with aims of WATER.
Answer: Thank you very much for your encouraging comments.
- 79-80: Please, add here more literature.
Answer: We added more literature based on your comment, as below:
“Noor et al. [26] indicated that all four remote-sensing rainfall methods significantly underestimate the rainfall intensities in Peninsular Malaysia throughout a range of durations and return periods. Nashwan et al. [29] compared the accuracy of five satellite precipitation products over Egypt and showed poor performance of all products in estimating rainfall over a predominantly arid climate region like Egypt. Ziarh et al. [30] showed bias in satellite precipitation varies with topography, could type and rainfall intensity.”
- Line 120: “IMERGE”? You mean IMERG?
Answer: Sorry for this mistake.
- Figure 3: In the flow chart you show some statistics that were not used in the study, for example the RMSE and the KGE. Or you should add in the statistical analysis these two statistics or correct the figure.
Answer: Sorry for this mistake. We modified the figure in the revised manuscript.
- Distribution functions: I do not understand why you limit so much the distributions of the analysis (GEV, Gumbel, GP)? There are other distributions that may fit better to your data. You rely on other studies, but I think that this is not appropriate.
For the selected distributions: It is well known that the length of the rainfall time series in very significant for the selection of the appropriate distribution (doi.org/10.1007/978-3-319-35095-0_48 and doi.org/10.1023/A:1008001312219). Your time series are very short (20 years) and this may have an impact on the selection of the best distribution. For that reason, you have to test other distributions like the EV-1. Add in the text (lines 155-170) an explanation using the above proposed literature.
Answer: Thank you for your comment. We added texts to explain using GEV, Gumbel and GP using the literature proposed by the reviewer. The added texts are as below:
“Koutsoyiannis and Baloutsos [45] reported that Extreme Value (EV1) and Gumbel seem more appropriate for a short period of data, while GEV may be better for esti-mating a larger return period. Kastridis and Stathis [46] also showed that the length of the rainfall time series significantly influences the selection of the appropriate distri-bution. Considering the data period of 20 years used in this study, EV1 and Gumbel may be the more suitable. However, the present study relied on the findings of the pre-vious studies in Iraq to select the distributions for their relative comparisons and iden-tify the best distribution.”
- Lines 323-329: This paragraph is a repetition of the Introduction. I think that is not necessary and you can remove it.
Answer: We deleted this paragraph based on your comment.

Round 2
Reviewer 1 Report
The manuscript entitled Utilizing Satellite Data to Establish Rainfall Intensity-Duration-Frequency Curves for Major Cities in Iraq can be accepted. The authors have addressed all my comments and corrected all errors in the text.
In my opinion, the manuscript can be accepted in its present form.
Sincerely,
The reviewer #3
Author Response
The manuscript entitled Utilizing Satellite Data to Establish Rainfall Intensity-Duration-Frequency Curves for Major Cities in Iraq can be accepted. The authors have addressed all my comments and corrected all errors in the text.
In my opinion, the manuscript can be accepted in its present form.
Answer: Thank you for your comment.
Reviewer 2 Report
Dear authors
Thank you very much for the provided responses to my comments. You have answered to all my comments and suggestions with a plausible way. I believe that the paper has significantly improved. However, I have two more comments. In lines 87-88 (revised version), please, also add the two proposed studies (doi.org/10.30955/gnj.003905 and doi: 10.3390/RS12091426), which show that the underestimation of extreme rainfall events is a serious problem of the satellite data and have as a result the wrong estimation of flood hydrographs. Also, in lines, 33-35 (revised version), I think that you should add a phrase about the anthropogenic impacts on the hydraulic dimensions of the rivers and streams, which are significantly decreased in urban areas due to the intensive urban sprawl. Good luck.
Author Response
Thank you very much for the provided responses to my comments. You have answered to all my comments and suggestions with a plausible way. I believe that the paper has significantly improved. However, I have two more comments.
Answer: Thank you very much for your comments. We revised the paper based on your comments.
- In lines 87-88 (revised version), please, also add the two proposed studies (doi.org/10.30955/gnj.003905 and doi: 10.3390/RS12091426), which show that the underestimation of extreme rainfall events is a serious problem of the satellite data and have as a result the wrong estimation of flood hydrographs.
Answer: We added the mentioned references based on your comment.
- Also, in lines, 33-35 (revised version), I think that you should add a phrase about the anthropogenic impacts on the hydraulic dimensions of the rivers and streams, which are significantly decreased in urban areas due to the intensive urban sprawl.
Answer: We added the phrase based on your account as below:
“Urbanization causes a reduction of permeable surfaces and increased runoff, which eventually alters the urban hydraulic dynamics and the dimension of rivers and streams [2]. This contributes to the increased frequency and severity of urban floods.”